# Safely managed sanitation practice and childhood stunting among under five years old children in Myanmar

**Than Kyaw Soe**⊙[ID], **Wongsa Laohasiriwong**⊙, **Kittipong Sornlorm**[ID]*⊙, **Roshan Kumar Mahato**⊙

Faculty of Public Health, Khon Kaen University, Khon Kaen, Thailand

⊙ These authors contributed equally to this work.
* kittsorn@kku.ac.th

**Data Availability Statement:** The data has been deposited in DOI-provided data repository. URL: https://datadryad.org/stash/share/2PvdInZW-

## Abstract

In 2020, 149 million children under the age of five were estimated to be stunted globally. Around half of deaths among children under 5 years of age are related to under-nutrition. Objective of this study is to determine the association between safely managed sanitation and childhood stunting among under-five years old children in Myanmar. This cross-sectional analytical study was conducted in 16 townships across three regions and five states in Myanmar. Multiple logistic regressions analysis was performed to determine the associations. This study found that 327 (27.09%) under-five children were stunted among a total of 1207 children in Myanmar. Children with unsafely managed sanitation were 2.88 times more likely to be stunting compared with children who access to safely managed sanitation services (AOR = 2.88, 95% CI: 2.16 to 3.85; p-value <0.01). Other associated factors for childhood stunting were needs 1–15 minutes for water collection (AOR = 2.07, 95% CI: 1.46 to 2.94; p-value <0.01), 15–60 minutes for water collection times (AOR = 1.55, 95% CI: 1.08 to 2.23; p-value 0.02), improper waste water disposal (AOR = 1.99, 95% CI: 1.47 to 2.70; p-value <0.01), boys children (AOR = 4.49, 95% CI: 3.30 to 6.12; p-value <0.01), did not take vitamin A supplements(AOR = 1.64, 95% CI: 1.22 to 2.20; p-value <0.01), mothers height shorter than 153.4cm (AOR = 1.94, 95% CI: 1.45 to 2.58; p-value <0.01), and the lower minimal diet diversity (AOR = 1.47, 95% CI: 1.08 to 2.01; p-value 0.02). More access to safely managed sanitation facilities, technical sharing for proper waste water disposal, promoting household water supply system, health promotion for children's diet eating pattern, and regular support for Vitamin A supplementation are critical to reduce childhood stunting among children under the age of five in Myanmar.

## 1. Introduction

In 2020, 149 million children under the age of five were stunted globally. Undernutrition is responsible for more than half of all fatalities in children under the age of five [1]. Moreover, half of all stunted children under the age of five lived in Asia and Africa [2]. Although global

nGe5WYZ_lq1vfNeZC6DL1nShFSibf9o1B8 DOI:
10.5061/dryad.76hdr7t2w.

**Funding:** The author(s) received no specific funding for this work.

**Competing interests:** The authors have declared that no competing interests exist.

malnutrition has declined gradually since 2000, rapid progress is required to meet the 2030 SDG target.

Malnutrition is a major risk factor for childhood death because it attracts infectious diseases, increases the frequency and severity of illness, and impedes disease recovery [3]. If a child is malnourished, he or she has a low chance of achieving modest scholastic and intellectual success in adulthood, as well as low production, a falling social level, and generational difficulties. Furthermore, malnutrition is linked to a higher risk of school dropout, lower socioeconomic status, and destitution in the long term [4, 5].

Chronic malnutrition causes frequent bouts of acute diarrhea in children. Furthermore, diarrhea remains a leading cause of death among children under the age of five, accounting for nearly 8% of all deaths among children under the age of five worldwide in 2017. Infectious diseases such as cholera, diarrhea, dysentery, hepatitis A, typhoid, and polio are spread due to poor sanitation. It is estimated that over 280,000 people die each year from diarrhea due to inadequate sanitation [6].

In Myanmar, diseases connected to water, sanitation, and hygiene (WASH) are common. In the long run, the nutritional status of children under the age of five is poor when compared to other regional countries. Furthermore, 10% of children under the age of five had diarrhea, and 3% had acute respiratory infection symptoms in the two weeks preceding the data collection period in the 2015–2016 Myanmar Demographic and Health Survey [7]. According to UNICEF, WHO, and the World Bank's joint child malnutrition estimate (JME) for 2022, 24.1% of children under the age of five are stunted, 7.4% are wasting, and 0.8% are overweight in Myanmar [8].

Although every country strives to meet the SDG target of safely managed sanitation by 2030, there is no dedicated survey to examine the level of safely managed sanitation status in Myanmar. Though various studies have been undertaken to verify the relationship between sanitation behavior and malnutrition among underfive children, limited studies have been conducted to investigate the relationship between access to safely managed sanitation and stunting among underfive children in Myanmar. As a consequence, this research study will be useful for future studies in Myanmar and other countries relating to underfive children's nutrition. The evidence-based recommendations in this study were useful to policymakers in planning sanitation and nutrition promotion for childhood malnutrition reduction programs.

## 2. Materials and methods

### 2.1 Study design and participants

This study was carried out in 16 townships across three regions and five states in Myanmar. To cover all parts of Myanmar, Kachin, Kayin, Mon, Rakhine, Shan states, and Magway, Mandalay, and Ayeyarwady regions were included in this study. A cross-sectional analytical study was carried out to determine the association between safely managed sanitation practices and stunting in children under the age of five. The children in the study ranged in age from 6 to 59 months at the time of data collection.

The inclusion and exclusion criteria were met by the eligible sample participants. Inclusion participants were: i) children aged 6 months to less than 5 years old at the time of data collection; ii) children whose parents granted informed consent to participate in this study; and iii) children who had lived in this area for at least a year. Exclusion criteria included: i) a child suffering from major health problems such as cancer; HIV, etc.; ii) a child with congenital disorders; iii) a child with any physical or mental disability; and iv) a child living with a single mother. The multiple logistic regression formula was used to calculate the sample size (Hsieh,

Bloch, & Larsen, 1998). According to the calculation result of this formula, 1207 children aged under five were the sample size of this study.

Samples were chosen using multistage sampling. First, 16 townships from eight targeted states and regions were randomly selected. Then, using simple random sampling, 1 ward was chosen from the urban context, and 2 villages for the rural setting were chosen from these townships. According to the ward composition of Myanmar, 1 ward represents urban characteristics such as coexistence of poor, rich, educated, non-educated, accessing services, not accessing services, etc. However, for rural areas, information from one village could not reflect on the whole rural area. Therefore, 1 ward and 2 villages were selected for this study. And the household numbers, together with the names of the household leaders, were obtained from the township's year-end health count data. A sampling population frame was listed that met inclusion and exclusion criteria. Following that, 1207 sample houses with children under the age of five were chosen using a systematic random selection technique. Finally, heights of selected children, their mothers, and fathers were measured using conventional instruments, and their mothers were interviewed using preformed questionnaires. The questionnaire was developed by the researcher based on the existing literature review findings. The data was collected from September 27, 2022 to December 26, 2022.

## 2.2 Measures

Children's body height in centimeters (cm) was measured to the nearest 0.1 cm by using a metering object that was recognized and used in the township health department. According to the WHO definition, stunting is defined as a child's height being more than two standard deviations below the median height for children of the same age and sex [9].

According to the WHO/UNICEF Joint Monitoring Program (JMP) definition, safely managed sanitation was defined as practicing the improved sanitation facilities by not sharing with other families in which excreta were disposed safely in situ or removed and providing treatment outside. At least basic drinking water was classified as drinking water from an improved source, provided collection time is not more than 30 minutes for a round trip, including queuing. The basic hygiene level was defined as the availability of a handwashing facility on premises with soap and water [10].

Children using toilets, rinsing feces into toilets, and buried practices were considered proper child feces disposal, but rinsing feces into drains, throwing them into garbage, leaving them open, and using them as manuals were considered improper child feces disposal. Disposal of garbage via formal or informal service provider, disposal in designated locations, burying, and burning were described as proper solid waste disposal, whereas disposing within one's own plot, disposing elsewhere, and don't know were regarded as improper solid waste disposal. Drains to the sewer line, designated pit, and soaking pit were considered proper waste water disposal, whereas drains to septic tanks, open ground, water bodies, and elsewhere were considered improper waste water disposal.

Health literacy was assessed in the areas of sanitation promotion, hygiene promotion, and nutritional promotion. Health literacy score intervals were classified into four levels for each percentage: less than 60 percent as "inadequate," 60 to 69 percent as "problematic," 70 to 79 percent as "adequate," and more than 80 percent as "excellent."

There were seven parts to the question structure: 1) sociodemographic factors, 2) child factors; 3) parent factors; 4) WASH factors; 5) food safety and diet pattern; 6) health literacy; and 7) disease factors. In sociodemographic factors, place of residence, kitchen location, having animal feces in the house, house floor type, and family size were included. Age, gender, parity, gestational age, birth weight, birth interval, Vitamin A supplement, deworming, antenatal care

visit and exclusive breast feeding were included in child factor questionnaires. In terms of parent factors, age, height, education, occupation, income, smoking, nutritional knowledge, and attitude were added. Relating to WASH factors, service level of water supply, sanitation and hygiene, systematic hand washing of mother, father, and child, water collection time, water shortage, solid waste disposal, child feces disposal, and waste water disposal were included. Food safety score, main food, food diversity, and food avoidance information were included in food safety and diet pattern questions. Sanitation promotion, hygiene promotion, and nutrition health literacy were included for health literacy parts. And diarrhea, dysentery, worm infection and sickness times were questioned for disease factors.

## 2.3 Statistical analysis

Microsoft Excel was used to record the raw data of 1207 respondents. The data were inverted using Stata version 13.1. For categorical data, the participants' socio-demographic and baseline characteristics were characterized using frequency and percentage, and for continuous data, mean, median, minimum, maximum, and standard deviation were presented. Multiple logistic regression, adjusted OR with a 95% confidence interval, and p-value were used to determine the association between safely managed sanitation and stunting. All test data were two-sided, and statistical significance was defined as a p-value less than 0.05.

## 2.4 Ethical consideration and consent to participate

This study protocol was approved by the Khon Kaen University Ethics Committee for Human Research with the reference number HE652157. Informed consents was obtained from the mothers and fathers of the children before interviewing sessions.

# 3. Results

## 3.1 Prevalence of stunting among under five children

Stunting prevalence by age and gender among underfive children in Myanmar was presented in Table 1. In a total of 1207 under-five children, 327 (27.09%) were stunted. The mean age Z-value (HAZ) for height was -1.10 (minimum -4.38, maximum 3.34). Stunting prevalence by age group is highest in children aged 2 to 3 years (33.62%), followed by children aged 4 to 5 years (32.38%), 6 months to 1 year (28.38%), 1 to 2 years (26.46%), and 3 to 4 years (15.69%). In terms of the prevalence of stunting by gender, 14.64 percent of all girls and 38.13 percent of all boys were stunted.

## 3.2 Baseline characteristics of under five years old children

Table 2 shows the baseline characteristics of the respondents. The majority of them were from rural areas (78.62%). 84.59 percent of children's kitchens are located inside the home. The majority of respondents' house floors were made of concrete or wood. Two-thirds of families had less than five members. In this survey, the boy-to-girl ratio was 47:53. Eighty percent of families had one or two children. 3.35 percent of babies were born prematurely, while 5.05 percent were born with a low birth weight. Two-thirds of the children took Oral vitamin A supplements. 59.15 percent of all children received deworming medications, and 78.13 percent had antenatal treatment while pregnant. The majority of children (76.22%) was treated with exclusive breast feeding.

43.58 percent of mothers were under 30 years of age, 46.31 percent were 30 to 40 years old, and 10.11 percent were over 40 years old. The median height of the mother was 154 cm, and the median height of the father was 158 cm. One out of ten mothers graduated. 10.11 percent

**Table 1. Prevalence of stunting by their age and gender among under-five children in Myanmar.**

| Characteristics | Total | Stunting | | 95% (CI |
|---|---|---|---|---|
| | | Number (n) | Percentage (%) | |
| **Gender** | | | | |
| Girls | 567 | 83 | 14.64 | 11.96 to 17.80 |
| Boys | 640 | 244 | 38.13 | 34.43 to 41.96 |
| **Age** | | | | |
| 6 months to 1 year | 222 | 63 | 28.38 | 22.82 to 34.68 |
| 1 to 2 years | 257 | 68 | 26.46 | 21.41 to 32.21 |
| 2 to 3 years | 229 | 77 | 33.62 | 27.79 to 40.01 |
| 3 to 4 years | 255 | 40 | 15.69 | 11.71 to 20.70 |
| 4 to 5 years | 244 | 79 | 32.38 | 26.79 to 38.52 |
| **Total** | **1207** | **327** | **27.09** | **24.66 to 29.67** |

of mothers and 8.62 percent of fathers worked as public or private employees. 55.51 percent earned less than 100 USD in income per month. 2.07 percent of mothers and 43.08 percent of fathers were current smokers. 94.70 percent of moms had good nutritional knowledge, and 77.22 percent of mothers had a positive attitude toward nutrition.

In terms of water, sanitation, and hygiene (WASH) characteristics, 68.43 percent were accessible for at least basic water supply services. Six out of ten families had safely managed sanitation services. Nine out of ten families had a basic hygiene level. 56.09 percent of all households do not require time to acquire water and can obtain it at home. Last year, 5.97 percent of people had a water deficit. 15.08 percent of households improperly dispose of child feces. Three out of every ten houses improperly dispose of garbage, and 55.92 percent of households improperly dispose of waste water.

The food security score ranged from 12 to 24, with 20 being the mean and median. Rice was the major diet for nine out of 10 children. 41.51 percent of children consumed at least four different types of foods. 19.20 percent of all children avoided some kinds of food. The majority of children's mothers had appropriate health literacy scores, with 53.11 percent having adequate sanitation health literacy, 43.58 percent having adequate hygiene promotion health literacy, and 51.78 percent having adequate nutrition promotion health literacy. Before two weeks, 7.13 percent had diarrhea, 4.23 percent had dysentery, and 3.98 percent had a worm infection. In the previous month, 46.06 percent of children were reported as unwell.

### 3.3 Factors associated with stunting among under five years old children

Bivariate analysis found that house floor type, family size, gender of child, gestational age, vitamin A supplement, antenatal care, exclusive breast feeding, maternal height, paternal height, mother's education status, family income, mother smoking, father smoking, maternal attitude on nutrition, water services level, water collection time, water shortage experience, child feces disposal, solid waste disposal, waste water disposal, food safety score, minimum diet diversity, sanitation promotion health literacy, nutrition promotion health literacy, diarrhea, dysentery, worm infection, and sickness were less than 0.25 p-value. Therefore, these variables were selected to add to the multivariable analysis model.

Table 3 present factors associated with stunting among under five years old children in Myanmar. After controlling the confounding factors with backward elimination multivariable analysis, safely managed sanitation was found to be a strongly significant factor for childhood stunting among under-five children. In terms of other WASH factors, water collection times

**Table 2. Baseline characteristics of the respondents.**

| Characteristics | Total (n = 1207) | |
|---|---|---|
| | Number | Percentage |
| **Place of residence** | | |
| Urban | 258 | 21.38 |
| Rural | 949 | 78.62 |
| **Kitchen location** | | |
| Outside home | 186 | 15.41 |
| Inside home | 1021 | 84.59 |
| **House floor type** | | |
| Concrete or wood | 1023 | 84.76 |
| Earth or bamboo | 184 | 15.24 |
| **Family size** | | |
| < 5 members | 853 | 70.67 |
| ≥ 5 members | 354 | 29.33 |
| Mean (SD) | 4.08 | (± 1.17) |
| Median (Min:Max) | 4 | (3:11) |
| **Gender of child** | | |
| Boy | 567 | 46.98 |
| Girl | 640 | 53.02 |
| **Parity** | | |
| 1–2 children | 957 | 79.29 |
| 3 and more children | 250 | 20.71 |
| **Labor type** | | |
| Preterm labor (birth before 37 weeks) | 44 | 3.65% |
| Normal labor (birth after 37 weeks) | 1163 | 96.35 |
| **Vitamin A supplement** | | |
| No | 435 | 36.04 |
| Yes | 772 | 63.96 |
| **Deworming** | | |
| No | 493 | 40.85 |
| Yes | 714 | 59.15 |
| **Antenatal Care** | | |
| Yes | 943 | 78.13 |
| No | 264 | 21.87 |
| **Exclusive breast feeding** | | |
| No | 287 | 23.78 |
| Yes | 920 | 76.22 |
| **Mother's current age** | | |
| Under 30 | 526 | 43.58 |
| 30–40 years | 559 | 46.31 |
| Over 40 years | 122 | 10.11 |
| **Mother's height in cm** | | |
| Less than 153.4 cm | 594 | 49.21 |
| Equal and more than 153.4 cm | 613 | 50.79 |
| Mean (S.D.) | 154.19 | (± 5.11) |
| Median (Min: Max) | 154 | (145: 195) |
| **Father's height in cm** | | |
| Less than 163.5 cm | 947 | 78.46 |

(*Continued*)

**Table 2.** (Continued)

| Characteristics | Total (n = 1207) | |
|---|---|---|
| | **Number** | **Percentage** |
| Equal and more than 163.5 cm | 260 | 21.54 |
| Mean (S.D.) | 159.26 | (± 5.03) |
| Median (Min: Max) | 158 | (150: 178) |
| **Mother's education** | | |
| Graduated | 122 | 10.11 |
| Non graduated | 1085 | 89.89 |
| **Monthly family income in USD** | | |
| Less than 100 USD | 670 | 55.51 |
| Equal and more than 200 USD | 537 | 44.49 |
| Mean (SD) | 118.50 | (± 77.55) |
| Median (Min:Max) | 95 | (2: 1286) |
| **Mother smoking** | | |
| No | 1182 | 97.93 |
| Yes | 25 | 2.07 |
| **Father smoking** | | |
| No | 687 | 56.92 |
| Yes | 520 | 43.08 |
| **Mother's knowledge on child malnutrition** | | |
| Good | 1143 | 94.70 |
| Poor | 64 | 5.30 |
| **Mother's attitude on child malnutrition** | | |
| Good | 932 | 77.22 |
| Poor | 275 | 22.78 |
| **Drinking water services level** | | |
| At least basic | 826 | 68.43 |
| Limited | 14 | 1.16 |
| Unimproved | 134 | 11.10 |
| Surface water | 233 | 19.30 |
| **Sanitation services level** | | |
| Safely managed | 719 | 59.57 |
| Basic | 80 | 6.63 |
| Limited | 221 | 18.31 |
| Unimproved | 100 | 8.29 |
| Open defecation | 87 | 7.21 |
| **Hygiene services level** | | |
| Basic | 1058 | 87.66 |
| Limited | 49 | 4.06 |
| No handwashing facility | 100 | 8.29 |
| **Water collection time** | | |
| No need to go for water collection | 677 | 56.09 |
| 0 to 15 minutes | 302 | 25.02 |
| 15 to 60 minutes | 228 | 18.89 |
| Median (Min: Max) | 18 | (2:29) |
| Median (Min: Max) | 0 | (0:60) |
| **Water shortage experience** | | |
| No | 1135 | 94.03 |

(*Continued*)

**Table 2.** (Continued)

| Characteristics | Total (n = 1207) | |
|---|---|---|
| | **Number** | **Percentage** |
| Yes | 72 | 5.97 |
| **Child feces disposal** | | |
| Proper disposal | 1025 | 84.92 |
| Improper disposal | 182 | 15.08 |
| **Solid waste disposal** | | |
| Proper disposal | 836 | 69.26 |
| improper disposal | 371 | 30.74 |
| **Waste water disposal** | | |
| Proper disposal | 532 | 44.08 |
| Improper disposal | 675 | 55.92 |
| **Overall food safety score** | | |
| Less than 17 | 99 | 8.20 |
| 17 to 24 | 1108 | 91.80 |
| Mean (SD) | 19.67 | (± 2.18) |
| Median (Min:Max) | 20 | (12:24) |
| **Main food** | | |
| Rice and others | 1,110 | 91.96 |
| Breastmilk | 97 | 8.04 |
| **Minimum dietary diversity at least 4** | | |
| Yes | 706 | 58.49 |
| No | 501 | 41.51 |
| **Eating times per day** | | |
| Less than 4 times | 850 | 70.42 |
| Equal and more than 4 times | 357 | 29.58 |
| **Food avoiding** | | |
| No | 974 | 80.70 |
| Yes | 233 | 19.30 |
| **Sanitation promotion health literacy** | | |
| Inadequate | 217 | 17.98 |
| Problematic | 263 | 21.79 |
| Adequate | 641 | 53.11 |
| Excellent | 86 | 7.13 |
| **Hygiene promotion health literacy** | | |
| Inadequate | 326 | 27.01 |
| Problematic | 283 | 23.45 |
| Adequate | 526 | 43.58 |
| Excellent | 72 | 5.97 |
| **Nutrition promotion health literacy** | | |
| Inadequate | 211 | 17.48 |
| Problematic | 298 | 24.69 |
| Adequate | 625 | 51.78 |
| Excellent | 73 | 6.05 |
| **Diarrhea in past two weeks** | | |
| No | 1121 | 92.87 |
| Yes | 86 | 7.13 |
| **Dysentery in past two weeks** | | |

(*Continued*)

**Table 2.** (Continued)

| Characteristics | Total (n = 1207) | |
| --- | --- | --- |
| | Number | Percentage |
| No | 1156 | 95.77 |
| Yes | 51 | 4.23 |
| **Worm infection in past two weeks** | | |
| No | 1159 | 96.02 |
| Yes | 48 | 3.98 |
| **Sickness in last month** | | |
| No | 651 | 53.94 |
| Yes | 556 | 46.06 |

and waste water disposal were found to be associated factors for stunting among under five years old children in Myanmar.

Children who accessed unsafely managed sanitation services, meaning that they practiced basic, limited, unimproved, and open defecation, were more likely to be stunted compared with children who accessed safely managed sanitation services about three times (AOR = 2.88, 95% CI: 2.16 to 3.85; p-value <0.01).

Moreover, those children whose families need to take 1 to 15 minutes for water collection (AOR = 2.07, 95% CI: 1.46 to 2.94; p-value <0.01) were significantly more likely to be stunted two times than the children whose families do not need to take time for water collection. Likewise, children whose families need to take more than 15 to 60 minutes for water collection had

**Table 3.** Factors associated with stunting among under five years old children in Myanmar.

| Variable | Total (No.) | Stunting (%) | Crude OR | Adjusted OR | 95% CI | p-value |
| --- | --- | --- | --- | --- | --- | --- |
| **Sanitation services level** | | | | | | <0.01 |
| Safely managed sanitation | 719 | 19.89 | 1 | 1 | | |
| Non-safely managed sanitation | 488 | 37.70 | 2.44 | 2.88 | 2.16 to 3.85 | |
| **Water collection time** | | | | | | <0.01 |
| No need time | 677 | 21.12 | 1 | 1 | | |
| 1–15 minutes | 302 | 35.43 | 1.98 | 2.07 | 1.46 to 2.94 | |
| 15–60 minutes | 228 | 33.77 | 1.83 | 1.55 | 1.08 to 2.23 | |
| **Waste water disposal** | | | | | | <0.01 |
| Proper disposal | 532 | 18.42 | 1 | 1 | | |
| Improper disposal | 675 | 33.93 | 2.23 | 1.99 | 1.47 to 2.70 | |
| **Gender of child** | | | | | | <0.01 |
| Girl | 567 | 14.64 | 1 | 1 | | |
| Boy | 640 | 38.13 | 3.59 | 4.49 | 3.30 to 6.12 | |
| **Vitamin A supplement** | | | | | | <0.01 |
| Yes | 435 | 22.54 | 1 | 1 | | |
| No | 772 | 35.17 | 1.86 | 1.64 | 1.22 to 2.20 | |
| **Mother's height in cm** | | | | | | <0.01 |
| Equal and more than 153.4 cm | 613 | 22.19 | 1 | 1 | | |
| Less than 153.4 cm | 594 | 32.15 | 1.66 | 1.94 | 1.45 to 2.58 | |
| **Diet diversity at least 4** | | | | | | 0.02 |
| Yes | 706 | 22.95 | 1 | 1 | | |
| No | 501 | 30.03 | 1.44 | 1.47 | 1.08 to 2.01 | |

55% more chances of being stunted than reference children (AOR = 1.55, 95% CI: 1.08 to 2.23; p-value 0.02). Furthermore, children who live in a family with improper waste water disposal practices were two times more likely to be stunted than the children with a family practicing proper waste water disposal methods.

Children's gender, taking vitamin A supplements, mother's height, and minimum diet diversity were also associated factors in childhood stunting. Under five years old boys were 4.5 times more likely to be stunted than the girls aged under five (AOR = 4.49, 95% CI: 3.30 to 6.12; p-value <0.01). Besides, children who did not take vitamin A supplements had 64% more chances of being stunted than other children who took vitamin A supplements (AOR = 1.64, 95% CI: 1.22 to 2.20; p-value <0.01). Moreover, children who bore from mothers with shorter than 153.4 cm were more likely to be stunted about 2 times than the children from mothers with taller than 153.4 cm (AOR = 1.94, 95% CI: 1.45 to 2.58; p-value <0.01). Furthermore, children who did not eat a minimum diversity of diet at least 4 items were significantly stunted, about 50% more than children who ate at least 4 diversities of foods.

## 4. Discussion

Childhood malnutrition is a critical health problem in Myanmar, which is one of the most malnourished countries in Southeast Asia. This study found that 327 (27.09%) of 1207 children were stunted, which means that nearly three out of every ten children under the age of five were shorter than their median height for the age of normal children. The prevalence of stunting in this study was slightly lower than the Myanmar Demographic Health Survey 2016 (MDHS 2016) result of 29 percent [7].

Safely managed sanitation practice was significantly associated with childhood stunting among under-five children in Myanmar after controlling for confounding factors with backward elimination multivariable analysis. Other two water, sanitation, and hygiene (WASH) variables such as water collection times, waste water disposal, and other four child-related factors such as children's gender, taking vitamin A supplements, mother's height, and minimum diet diversity were also associated with childhood stunting among children under the age of five in Myanmar.

Children who had access to safely managed sanitation services were three times less likely to be stunted than other children. The conclusions of this study supported the findings of the previous Indonesia study [11]. An Indonesian study presented that ambient sanitation characteristics where children live, as well as ownership of a semi-permanent toilet, have a strong association with the occurrence of stunting. However, this new finding of an association between safely managed sanitation and stunting supports the prior systematic review and meta-analysis findings of Rosiyati, Vilcin, and Larsen about the relationship between childhood stunting and sanitation status [12–14].

The Momberg study on South Africa explored a relationship between safely managed sanitation and children's nutritional health [15] of under one year old children. The current study findings support the previous conclusions on the association between safely managed sanitation and childhood stunting. Furthermore, this new finding on sanitation service level and childhood stunting contributed to the findings of India [16], Pakistan [17], Indonesia [18] and Ethiopia [19] on this association of sanitation and childhood stunting. Therefore, we need to focus on improving safely managed sanitation facilities in the community to decrease child malnutrition.

Difficulty getting water and the time needed for water collection were also risk factors for childhood stunting. This study identified that under five years old children from the home with no need to pay time for water collection can reduce stunting from 8% to 3 times. A previous study [20] also found that two- to five-year-old children from households with improved

water sources and pipe water systems had a lower opportunity to be stunted. Water can be contaminated throughout the water collection route, such as the water source, fetching, carrying, carrier material, etc. This finding supports the former study of Mashida, who highlighted that children who use domestic water from surface water sources such as rivers, streams, etc., were nine times more likely to be malnourished than those from households with pipe water [21]. Therefore, all these findings highlighted the importance of household water accessible at home for child nutrition improvement.

This study explored that improper waste water disposal was the predisposing factor for childhood stunting. Children from a family with proper waste water disposal practice were two times less likely to be stunted than the children with improper waste water disposal practices. Waste water contains dissolved matter and microorganisms that may be harmful to humans, animals, and the environment. Unproper waterways and poor drainage lines were the sources for fly and mosquito breeding, which can cause various infections. Frequent sickness and infection affect childhood nutritional status. There were no previous studies on the relationship between waste water disposal type and childhood nutritional status.

Gender of children was the associated factor for childhood stunting, as proved by many studies. This study found that boys were more likely to be stunted by about 4.5 times than girls in Myanmar. This study result was the same as the findings of Tanzania [20], Nigeria [22], and Indonesia [23] studies; those studies found that male children were more likely to be stunted than females. However, the odds of being stunted were about three times higher for Ethiopian females [24]. Indian girls had higher heights for age Z scores than boys [25]. This finding fulfilled previous Myanmar literature showing the association between gender and childhood stunting. That study revealed that boys had a higher risk of being stunted than girls in Myanmar [26].

This study found that children who took vitamin A supplements were less likely to be stunted than other children. The Uganda study also evidenced that vitamin A deficiency was the predisposing factor for childhood stunting [27]. Similarly, one Iranian national study identified an association between serum retinol levels and children's stunting [28]. This finding contributed to the previous findings on the relationship between vitamin A supplements and child stunting. Regular vitamin A supplements for under five children play an important role in decreasing stunting in children.

Previous Myanmar studies by Kang and Phyo proved that children from short mothers had 2.5 to nearly 4 times more chances of being stunted than the children from tall mothers [26, 29]. Moreover, the Li systematic review paper finding supports this finding that a short maternal height has four times more chances of having a stunted child than a tall [30]. This study fulfilled the previous findings about this relationship. This study explored that children whose mother was shorter than Asians' average woman heights were more likely to be stunted, about twice as likely as the children from taller mothers.

This study also found that eating a diverse variety of food was the most significant factor determining stunting in under-five children. Children eating a minimum diet of at least 4 items were less likely-about 53%-than children who eat diverse foods less than 4 items per day. A similar result was evidenced in one India study, which presented that lower consumption of varieties of foods such as grains, roots, and tubers was associated with a 34% higher risk of childhood stunting [16]. These studies encouraged all 6-59-month-old children to eat at least four varieties of food per day for their nutritional improvement.

## 4.1 Strengths of study

This study explored the prevalence of stunting among children under five years old in Myanmar. It is the first report on the association between safely managed sanitation practices and

under-five children's stunting in Myanmar. Therefore, this research study can be a reference for similar future studies that will be conducted in Myanmar and also in different countries. Moreover, this study would be useful for policymakers in drawing up a sanitation and nutrition development program for malnutrition reduction among under five years old children.

### 4.2 Limitations of study

Since the current study was a cross-sectional analytical study, further study with operational research or a longitudinal cohort study design was recommended to provide a better understanding of the relationship between safely managed sanitation practices and childhood malnutrition.

## 5. Conclusion

There was a high prevalence of stunting among children aged 5 to 59 months old in Myanmar. Children who were accessible to unsafely managed sanitation facilities were strongly associated with childhood stunting among under-five children in Myanmar. And improper waste water disposal practices and the time needed to collect drinking water were the WASH-related influencing factors for stunting among under-five children. Moreover, eating at least four diverse food items, maternal height, gender of children, and vitamin A supplements were also significant factors in determining childhood stunting.

The following recommendations were made to enhance the nutritional status of children under the age of five: i) to promote the accessibility of the community's safely managed sanitation facilities; ii) to provide technical training and knowledge sharing sessions to the community about household waste water management; iii) to promote the availability of drinking water at home; iv) to provide health education to mothers for the minimum diet diversity needs of under five children; v) to regularly provide Vitamin A supplements to 6-59-month-old children; and vi) to conduct the further study with operational research or a longitudinal cohort study design for understanding more of the relationship between safely managed sanitation practices and childhood malnutrition.

## Supporting information

**S1 File. Questionnaires of "safely managed sanitation practice and childhood stunting among under five years old children in Myanmar".**
(DOCX)

## Acknowledgments

The authors would like to express their sincere thanks to village and ward administrators, community leaders, and health staff of the study Townships for helping in data collection. And the author would like to thank all mothers and fathers of under-five children in the study area for their participation and their commitment.

## Author Contributions

**Conceptualization:** Than Kyaw Soe, Kittipong Sornlorm.

**Formal analysis:** Than Kyaw Soe, Kittipong Sornlorm, Roshan Kumar Mahato.

**Investigation:** Than Kyaw Soe.

**Methodology:** Than Kyaw Soe, Wongsa Laohasiriwong, Roshan Kumar Mahato.

**Supervision:** Wongsa Laohasiriwong, Kittipong Sornlorm, Roshan Kumar Mahato.

**Writing – original draft:** Than Kyaw Soe.

**Writing – review & editing:** Than Kyaw Soe, Wongsa Laohasiriwong, Kittipong Sornlorm.

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
