## [Decision Letter · Decision Letter 0]

3 Oct 2023

PONE-D-23-25415Safely managed sanitation practice and childhood stunting among under five years old children in MyanmarPLOS ONE

Dear Dr. Sornlorm,

Thank you for submitting your manuscript to PLOS ONE. After careful consideration, we feel that it has merit but does not fully meet PLOS ONE’s publication criteria as it currently stands. Therefore, we invite you to submit a revised version of the manuscript that addresses the points raised during the review process.

ACADEMIC EDITOR:

Dear author,

The paper has been reviewed by the esteemed reviewers. The paper needs through revision before reevaluating for possible publication. 

With regards,

Ranjit

We look forward to receiving your revised manuscript.

Kind regards,

Ranjit Kumar Dehury

Academic Editor

PLOS ONE

Additional Editor Comments:

Dear author,

The paper has been reviewed by the esteemed reviewers. The paper needs through revision before reevaluating for possible publication.

With regards,

Ranjit

Reviewers' comments:

Reviewer's Responses to Questions

**Comments to the Author**

1. Is the manuscript technically sound, and do the data support the conclusions?

Reviewer #1: Partly

Reviewer #2: Yes

2. Has the statistical analysis been performed appropriately and rigorously? 

Reviewer #1: No

Reviewer #2: Yes

3. Have the authors made all data underlying the findings in their manuscript fully available?

Reviewer #1: Yes

Reviewer #2: Yes

4. Is the manuscript presented in an intelligible fashion and written in standard English?

Reviewer #1: No

Reviewer #2: No

5. Review Comments to the Author

Reviewer #1: Title:

I suggest changing the title to "Prevalence and factors associated with stunting among children under five years in Myanmar. Safely managed sanitation practice is not the only variable found associated with stunting."

Methodology:

• The authors should clearly describe how they selected the 16 townships from eight states and regions. They should also explain the sampling frame and sampling strategy used. The justification for selecting one ward in urban areas and two villages in rural areas should also be explained.

• The authors should cite a reference for the definition of stunting. The WHO's definition is a good option: "Stunting is defined as a child's height being more than two standard deviations below the median height for children of the same age and sex."

• The authors should cite a reference for the WHO/UNICEF Joint Monitoring Program (JMP) definition of safely managed sanitation.

• The authors should define improved sources of water or cite the WHO/UNICEF standard. The WHO/UNICEF standard is: "Improved water supply means that people have access to an improved drinking water source, such as a piped water supply, borehole, protected well, or rainwater collection."

Results:

• Table 2 should be shortened by focusing on biodata and sociodemographic factors.

• The definitions of the variables in Table 2 should be clarified. For example, "adequate, inadequate, problematic promotion" should be defined.

• The authors should explain why they did not include all of the variables in Table 2 in the multivariable analysis. It is possible that the association between safe sanitary situation and stunting would be different if all of the variables were included. For robust results, all variables should be included in the multivariable analysis.

• The authors should clarify whether the 95% confidence interval (CI) in Table 3 is for the crude OR or the adjusted AOR: the AOR = 0.53, while 95%CI = 1.11, 1.87

Reviewer #2: The title of the paper is appropriate and best suited. The literature review and problem statement justified the need for the study.

The methodology of stratified sampling needs justification; why sixteen townships, one urban ward, and two rural areas?

Was the “preformed questionnaire” constructed by the authors or adopted? Please clarify.

Statistical analysis and results drawn were sound and supported the existing area of sanitation and stunting in children.

There are many grammatical and typographical errors in the text.

6. PLOS authors have the option to publish the peer review history of their article (what does this mean?). If published, this will include your full peer review and any attached files.

Reviewer #1: No

Reviewer #2: No

---

## [Author Response · Author response to Decision Letter 0]

9 Oct 2023

Editor comments:

Ans: Author contribution information has been added in title page to be align with PLOSONE structure guideline. And line numbers are added to manuscript file. Manuscript body meets with PLOS ONE's style requirements. 

2. We note that you have indicated that data from this study are available upon request. PLOS only allows data to be available upon request if there are legal or ethical restrictions on sharing data publicly. For more information on unacceptable data access restrictions, please see http://journals.plos.org/plosone/s/data-availability#loc-unacceptable-data-access-restrictions . 

Ans: The data has been deposited in DOI provided data repository. The link is below: https://datadryad.org/stash/share/2PvdlnZW-nGe5WYZ_Iq1vfNeZC6DL1nShFSibf9o1B8

Ans: there is no any ethical or legal restriction on sharing this data. 

Ans: The data has been deposited in DOI provided data repository. The link is below: https://datadryad.org/stash/share/2PvdlnZW-nGe5WYZ_Iq1vfNeZC6DL1nShFSibf9o1B8

Ans: Thank you so much for your support and kind understanding on that. 

Ans: Thank your for your suggestion. Ethical consideration and consent to participate statement has been moved to method section. And it has been removed from additional information part. 

4. Please include captions for your Supporting Information files at the end of your manuscript, and update any in-text citations to match accordingly. Please see our Supporting Information guidelines for more information: http://journals.plos.org/plosone/s/supporting-information . 

Ans: Caption has been revised to supporting documents to match with supporting information guideline. 

Reviewer 1 Comments:

I suggest changing the title to "Prevalence and factors associated with stunting among children under five years in Myanmar”. Safely managed sanitation practice is not the only variable found associated with stunting."

Ans: Yes, well noted that and thanks. Actually, this study focus to answer “Is there association between safely managed sanitation and underfive children stunting? Therefore, we put the main interest (x) and output (y) indicator on this study title. Other factors are covariates variables. 

Methodology:

• The authors should clearly describe how they selected the 16 townships from eight states and regions. They should also explain the sampling frame and sampling strategy used. The justification for selecting one ward in urban areas and two villages in rural areas should also be explained.

Ans: Township selection, sampling frame and ward/villages selection explanation has been revised in manuscript. 

• The authors should cite a reference for the definition of stunting. The WHO's definition is a good option: "Stunting is defined as a child's height being more than two standard deviations below the median height for children of the same age and sex."

Ans: Thank. It has been updated and added the reference.

• The authors should cite a reference for the WHO/UNICEF Joint Monitoring Program (JMP) definition of safely managed sanitation.

Ans: Reference has been added.

• The authors should define improved sources of water or cite the WHO/UNICEF standard. The WHO/UNICEF standard is: "Improved water supply means that people have access to an improved drinking water source, such as a piped water supply, borehole, protected well, or rainwater collection."

Ans: Reference has been added. 

Results:

• Table 2 should be shortened by focusing on biodata and sociodemographic factors.

Ans: Some factors (less focus) has been removed from table 2. 

• The definitions of the variables in Table 2 should be clarified. For example, "adequate, inadequate, problematic promotion" should be defined.

Ans: Health literacy scores were classified in method section. 

• The authors should explain why they did not include all of the variables in Table 2 in the multivariable analysis. It is possible that the association between safe sanitary situation and stunting would be different if all of the variables were included. For robust results, all variables should be included in the multivariable analysis.

Ans: According to the suggestion of statistician, the variables with crude odd ratio its pvalue more than 0.25 were not added on final maultivariate analysis model. I just revised to that paragraph to be more understanding to the reader. 

• The authors should clarify whether the 95% confidence interval (CI) in Table 3 is for the crude OR or the adjusted AOR: the AOR = 0.53, while 95%CI = 1.11, 1.87

Ans. Thank you so much. CI value in Table is for AOR. 0.53 it is my typing error. I have already corrected to that value. 

Reviewer 2 comments:

The title of the paper is appropriate and best suited. The literature review and problem statement justified the need for the study.

Ans: Thank for your positive comment and constructive feedback. 

The methodology of stratified sampling needs justification; why sixteen townships, one urban ward, and two rural areas?

Ans: Two townships of 8 states were selected with randomly. Ward/village section has been revised on the method section. 

Was the “preformed questionnaire” constructed by the authors or adopted? Please clarify.

Ans: It has been updated on Method section. 

Statistical analysis and results drawn were sound and supported the existing area of sanitation and stunting in children.

Ans: Thank for your positive remark.

There are many grammatical and typographical errors in the text.

Ans: Thank a lot for your constructive feedback and valuable comment. Grammer has been checked and corrected.

---

## [Decision Letter · Decision Letter 1]

6 Nov 2023

Safely managed sanitation practice and childhood stunting among under five years old children in Myanmar

PONE-D-23-25415R1

Dear Dr. Kittipong Sornlorm,

We’re pleased to inform you that your manuscript has been judged scientifically suitable for publication and will be formally accepted for publication once it meets all outstanding technical requirements.

Kind regards,

Ranjit Kumar Dehury

Academic Editor

PLOS ONE

Additional Editor Comments (optional):

Dear authors,

According to my reading and esteemed reviewers the article is found to be of acceptable quality. Hence, the article is accepted for publication.

With regards,

Ranjit

Reviewers' comments:

Reviewer's Responses to Questions

**Comments to the Author**

1. If the authors have adequately addressed your comments raised in a previous round of review and you feel that this manuscript is now acceptable for publication, you may indicate that here to bypass the “Comments to the Author” section, enter your conflict of interest statement in the “Confidential to Editor” section, and submit your "Accept" recommendation.

Reviewer #1: All comments have been addressed

Reviewer #2: All comments have been addressed

2. Is the manuscript technically sound, and do the data support the conclusions?

Reviewer #1: Yes

Reviewer #2: Yes

3. Has the statistical analysis been performed appropriately and rigorously? 

Reviewer #1: Yes

Reviewer #2: Yes

4. Have the authors made all data underlying the findings in their manuscript fully available?

Reviewer #1: Yes

Reviewer #2: Yes

5. Is the manuscript presented in an intelligible fashion and written in standard English?

Reviewer #1: No

Reviewer #2: Yes

6. Review Comments to the Author

Reviewer #1: (No Response)

Reviewer #2: (No Response)

7. PLOS authors have the option to publish the peer review history of their article (what does this mean?). If published, this will include your full peer review and any attached files.

Reviewer #1: No

Reviewer #2: No

---

## [Editor Report · Acceptance letter]

11 Nov 2023

PONE-D-23-25415R1 

Safely managed sanitation practice and childhood stunting among under five years old children in Myanmar 

Dear Dr. Sornlorm:

I'm pleased to inform you that your manuscript has been deemed suitable for publication in PLOS ONE. Congratulations! Your manuscript is now with our production department. 

Kind regards, 

on behalf of

Dr. Ranjit Kumar Dehury 

Academic Editor

PLOS ONE